

# High spatial resolution imaging of methane and other trace gases with the airborne Hyperspectral Thermal Emission Spectrometer (HyTES)

G. C. Hulley[1], R. M. Duren[1], F. M. Hopkins[1], S. J. Hook[1], N. Vance[1], P. Guillevic[2], W. R. Johnson[1], B. T. Eng[1], J. M. Mihaly[1], V. M. Jovanovic[1], S. L. Chazanoff[1], Z. K. Staniszewski[1], Le Kuai[1], John Worden[1], Christian Frankenberg[4], G. Rivera[1], A. D. Aubrey[1], C. E. Miller[1], N. K. Malakar[1], J. M. Sánchez Tomás[3], K. T. Holmes[1]

[1]Jet Propulsion Laboratory, California Institute of Technology, Pasadena, CA, USA, 91109
[2]Department of Geographical Sciences, University of Maryland, College Park, MD, 20742
[3]Universidad de Castilla-La Mancha, Ciudad Real, Spain
[4]California Institute of Technology, Pasadena, CA, USA, 91109

*Correspondence to*: G. C. Hulley (glynn.hulley@jpl.nasa.gov)

*(c) 2016 California Institute of Technology. Government sponsorship acknowledged*

**Abstract.** Currently large uncertainties exist associated with the attribution and quantification of fugitive emissions of criteria pollutants and greenhouse gases such as methane across large regions and key economic sectors. In this study, data from the airborne Hyperspectral Thermal Emission Spectrometer (HyTES) have been used to develop robust and reliable techniques for the detection and wide-area mapping of emission plumes of methane and other atmospheric trace gas species over challenging and diverse environmental conditions with high spatial resolution that permits direct attribution to sources. HyTES is a pushbroom imaging spectrometer with high spectral resolution (256 bands from 7.5- 12 μm), wide swath (1-2 km), and high spatial resolution (~2 m at 1 km altitude) that incorporates new thermal infrared (TIR) remote sensing technologies. In this study we introduce a hybrid Clutter Matched Filter (CMF) and plume dilation algorithm applied to HyTES observations to efficiently detect and characterize the spatial structures of individual plumes of $CH_4$, $H_2S$, $NH_3$, $NO_2$, and $SO_2$ emitters. The sensitivity and field of regard of HyTES allows rapid and frequent airborne surveys of large areas including facilities not readily accessible from the surface. The HyTES CMF algorithm produces plume intensity images of methane and other gases from strong emission sources. The combination of high spatial resolution and multi-species imaging capability provides source attribution in complex environments. The CMF-based detection of strong emission sources over large areas is a fast and powerful tool needed to focus more computationally intensive retrieval algorithms to quantify emissions with error estimates, and is useful for expediting mitigation efforts and addressing critical science questions.

## 1 Introduction

The Hyperspectral Thermal Emission Spectrometer (HyTES) is a pushbroom imaging spectrometer that produces a wide swath Thermal Infrared (TIR) image with high spectral (256 bands from 7.5- 12 μm) and spatial resolution (~2 m at 1 km altitude) (Hook et al., 2013; Hook et al., 2015). HyTES incorporates a number of technologies that presents a major advance in airborne TIR hyperspectral remote sensing measurements (Johnson et al., 2012; Johnson et al., 2009). While hyperspectral imaging spectrometers operating in the visible to shortwave spectrum (VSWIR, 1400-2500 nm), such as the Next Generation Airborne Visible Infrared Imaging Spectrometer (AVIRIS-NG) (Green et al., 1998), rely on reflected solar radiance to detect various chemical gas species such as methane ($CH_4$) (Roberts et al., 2010; Thompson et al., 2015; Thorpe et al., 2014; Thorpe et al., 2013),



TIR spectrometers instead rely on the thermal emission and thermal contrast between ground and target gas alone. This has the advantage of making detection more robust over a wider range of land cover types independent of their reflective features. For example, given sufficient thermal contrast between the plume and the surface, TIR data will have a distinct advantage over SWIR data for methane detection over low albedo surfaces such as seawater and dark vegetation, and particularly at higher latitudes

where reduced reflective solar insolation make it a challenge for current SWIR instrument capabilities. These kinds of conditions would be typical of the Arctic region, for example, which contains large reservoirs in the form of methane hydrates at the ocean surface and in permafrost regions (Damm et al., 2010; Kort et al., 2012). TIR observations also allow night-time operation during which the collapsed nocturnal planetary boundary layer results in higher near-surface concentrations of source gases – translating to easier detection.  Another key advantage of TIR hyperspectral data is the ability to distinguish between different trace gas

signatures within a single plume - a capability that will be demonstrated in this work.

TIR remote sensing has a long heritage of medium to high spatial resolution airborne and spaceborne sensors with multiple (3-10) bands in the TIR region starting with the 6-band Thermal-Infrared Multispectral Scanner (TIMS) airborne sensor in the early 1980's (Kahle and Goetz, 1983) and followed by the MODIS/ASTER (MASTER) airborne sensor with 10 bands in the TIR region (Hook et al., 2001). However, one of the biggest drawbacks of these imagers is their limited number of spectral bands

defining the TIR region (7.5 - 12 μm). In response, a number of hyperspectral TIR sensors have been developed, starting with the narrow field of view SEBASS (Spatially Enhanced Broadband Array Spectograph System) (Hackwell et al., 1996), and including wide-swath capabilities such as MAKO (Warren et al., 2010), the Mineral and Gas Identifier – in LEO (MAGI -L) (Hall et al., 2008), AisaOWL (Doneus et al., 2014), SIELETERS (Ferrec et al., 2014), and HyTES (Hook et al., 2013). Table 1 compares the instrument characteristics of each of these six sensors. Of these instruments, HyTES has the highest number of spectral bands

(256), which improves the detection sensitivity of trace atmospheric gas constituents such as $CH_4$ (Tratt et al., 2014) and ammonia ($NH_3$) (Tratt et al., 2011) by better discerning their residual spectral structure from ambient atmospheric absorption features. HyTES has sufficient spectral information in the 7.4-12 μm region to resolve the spectral absorption signatures of a variety of different trace gases including CH4, ammonia ($NH_3$), hydrogen sulfide ($H_2S$), sulfur dioxide ($SO_2$) and nitrogen dioxide ($NO_2$).

Airborne hyperspectral imagers such as HyTES have a wide-swath mapping capability and fine spatial resolution, making

them very useful for the detection of discrete sources of gaseous emissions over large regions, otherwise difficult from ground or airborne lidar measurements alone (Thorpe et al., 2014; Tratt et al., 2014). In the context of climate change and air quality, the ability to detect and characterize individual point sources of greenhouse gases such as methane or criteria pollutants such as sulfur and nitrogen oxides from key emitting sectors is a promising tool for improving understanding of the distribution of emissions sources and for supporting emissions mitigation.

In this work we present the theory and methodologies for the rapid detection of a variety of trace gas species ($CH_4$, $NO_2$, $NH_3$, $H_2S$, and $SO_2$) from the HyTES hyperspectral TIR data, with a focus on methane. We introduce a hybrid Clutter Matched Filter (CMF) and plume dilation algorithm for efficiently detecting and imaging trace gas plumes. We present representative results from field testing including the detection of anthropogenic $CH_4$ sources over challenging areas such as urban Los Angeles, where thermal in-scene clutter makes detection difficult, and over managed systems such as dairy farms, and oil fields in the San Joaquin

Valley (SJV), California. The sites were chosen to test the HyTES gas detection technique in a variety of different settings related to a range of science applications, and to provide in situ measurements to validate those results. The primary science goal of the HyTES flights over these sites was to detect, attribute, and characterize the spatial structure of $CH_4$ plumes to better understand their distribution and enable follow up measurements, and identify high priority sources for follow-up analysis with more computationally intensive quantitative retrievals (Kuai et al., 2015). We also demonstrate the ability of the technique to image

different chemical species such as $NH_3$, $H_2S$, $SO_2$ and $NO_2$ within the same plume. HyTES Level-1 radiance data and Level-2



Surface Temperature and Emissivity data from the 2013, 2014, and 2015 campaigns are free and available for ordering at http://www.hytes.jpl.nasa.gov/order.

**2. HyTES Background**

**2.1 Instrument**

5       The HyTES instrument is a Dyson optical configuration with a compact hyperspectral grating spectrometer acquiring data in 256 spectral bands in the TIR range from 7.5-12 µm (Figure 1), and a Quantum Well Infrared Photodetector (QWIP) (Gunapala et al., 2006). This is the first integration of the QWIP with a spectrometer imaging system for Earth science studies that require well calibrated data. A major advantage of the instrument is its very compact design due to its small form factor and relatively low power requirements (Johnson et al., 2012). A vacuum chamber is used to keep the focal plane system cold using two mechanical

cryocoolers (Figure 1). The chamber has also been proven to support airborne operation of other VSWIR instruments, while maintaining rigidity of its inner precision and optical components. A full description of the HyTES instrument including instrument performance, calibration and validation is provided by Hook et al (2015).

       A single sensor calibration is used for an entire field campaign with continuous data acquisition and geo-coding throughout. HyTES is currently configured to fly on the Twin Otter aircraft and Figure 1 shows the aircraft and the HyTES

instrument looking nadir in flight. For detection of trace gases, flights are usually conducted at an altitude of 1-km Above Ground Level (AGL) to minimize atmospheric attenuation between ground and sensor. The HyTES pixel size at 1-km AGL is ~2 m. Figure 2 shows an example of a HyTES data hypercube for a flight over Death Valley, California. Radiances in the vertical slice have been atmospherically corrected for the atmospheric transmission and path radiance using an in-scene atmospheric correction approach.

**3. Thermal Infrared Physics**

       The clear sky radiance measured by a sensor in the TIR spectral region (7−14 µm) is a combination of the Earth-emitted radiance, reflected downwelling sky irradiance, and atmospheric path radiance, and is defined as the flux per unit projected area per unit solid angle incident at the sensor. The Earth-emitted radiance is a function of the land surface temperature and spectral emissivity and gets attenuated by the atmosphere on its path to the sensor. The atmosphere also emits radiation, some of which

gets scattered up into the path of the sensor directly and called the atmospheric path radiance, while some gets radiated to the surface (irradiance) and reflected back to the sensor - termed the reflected downwelling sky irradiance. Reflected solar radiation in the TIR region is negligible and is not accounted for in forward simulations of at-sensor radiance. One effect of the sky irradiance is to reduce the spectral contrast of the emitted radiance, since the addition of the downward reflected component 'fills' in the spectral features from the surface.

**3.1 Theory**

       Using Kirchhoff's law we can write the hemispherical-directional reflectance as a function of directional emissivity ($\rho_\lambda = 1 - \epsilon_\lambda$), and express the at-sensor radiance for a clear-sky pixel with no gas plume attenuation ("off-plume"), ( $L_\lambda^{\text{off}}$ ) as follows:

$$L_\lambda^{\text{off}}(\theta) = L_\lambda^{\text{gnd}} \tau_\lambda^{atm} + L_\lambda^{\uparrow} \tag{1}$$



Where $\tau_\lambda^{atm}$ is the atmospheric transmittance, $L_\lambda^\uparrow$ is the atmospheric path radiance, and $L_\lambda^{gnd}$ is the total land leaving radiance:

$$L_\lambda^{gnd}(\theta) = L_\lambda^{surf} + \rho_\lambda L_\lambda^\downarrow = \epsilon_\lambda B_\lambda(T_s) + (1 - \epsilon_\lambda)L_\lambda^\downarrow \tag{2}$$

Where $\lambda$ = wavelength; $\theta$ = observation angle; $L_\lambda^{surf}$ = Earth-emitted radiance; $\epsilon_\lambda$ = spectral surface emissivity; $T_s$ = surface temperature; $L_\lambda^\downarrow$ = downwelling sky irradiance; $\tau_\lambda^{atm}$ = atmospheric transmittance; $L_\lambda^\uparrow$ = atmospheric path radiance; $B_\lambda(T_s)$ = Planck function defined at temperature, $T_s$.

The radiance measured by a sensor for a pixel centered on a gaseous plume ("on-plume") introduces an additional plume thermal emission term, $L_\lambda^p$, and a plume transmissivity term, $\tau_\lambda^p$ to account for the additional attenuation of the surface radiance.

$$L_\lambda^{on}(\theta) = L_\lambda^{gnd}\tau_\lambda^{atm}\tau_\lambda^p + L_\lambda^\uparrow + L_\lambda^p\tau_\lambda^{atm} \tag{3}$$

Where: $L_\lambda^p = \epsilon_\lambda^p B_\lambda(T_p)$ is the gas plume emission term, $T_p$ is the plume temperature, and $\epsilon_\lambda^p$ the plume emissivity. An illustration depicting these components for an observation over a gaseous plume is shown in Figure 3. These terms can be simplified using some physics-based assumptions. The weak plume transmissivity, $\tau_\lambda^p$, is given by Beer's law:

$$\tau_\lambda^p = e^{-n_o b_\lambda} \tag{4}$$

Where $n_o$ is the gas column density and $b_\lambda$ is the gas absorbance spectra ("plume signature"), usually extracted from the HITRAN database for the relevant gas constituent. If we assume the gas plume is optically thin and plume absorbance, $(n_o b_\lambda)$, is small, we can approximate (4) with a Taylor expansion:

$$\tau_\lambda^p \approx 1 - n_o b_\lambda \tag{5}$$

The Beer-Lambert law can then be used to write the transmittance as a function of the gas plume effective emissivity:

$$\epsilon_\lambda^p = 1 - \tau_\lambda^p \approx n_o b_\lambda \tag{6}$$

Substituting (6) into (3) and re-arranging terms yields an equation describing the total at-sensor radiance for an observation
centered on a gaseous plume pixel:

$$L_\lambda^{on}(\theta) = [L_\lambda^{gnd}\tau_\lambda^{atm} + L_\lambda^\uparrow] + n_o b_\lambda \tau_\lambda^{atm}[B_\lambda(T_p) - L_\lambda^{gnd}] \tag{7}$$

The first term on the right-hand-side of Eq. 7 describes the off-plume radiance, $L_\lambda^{off}$ (background, or 'clutter'), while the second term consists of the plume signature, $\boldsymbol{b}$, multiplied by the plume strength, which includes the column density, $n_o$, multiplied by the thermal contrast term, $[B_\lambda(T_p) - L_\lambda^{gnd}]$. Examination of this term indicates that detection of gaseous plumes in the TIR requires a strong thermal contrast between the surface and the plume, otherwise the plume strength term approaches zero. To solve Eq.7,
knowledge of the surface temperature of the background, the temperature of the gas plume, the surface emissivity of the background, and the atmospheric terms $\tau_\lambda^{atm}$ and $L_\lambda^\uparrow$ is needed. The atmospheric terms are estimated using an atmospheric correction technique described in the next section, and are usually fairly constant across an image at the few kilometers scale depending on variability in atmospheric water vapor.

**3.2 In-Scene Atmospheric Correction (ISAC) Methods**

The spectral radiance in (1) will include atmospheric emission, scattering, and absorption by the Earth's atmospheric constituents. In order to isolate the land-leaving surface radiance and separate the surface temperature and spectral emissivity terms, these atmospheric effects need to be removed from the observation. For on-plume pixel observations, the atmospheric



compensation isolates the land-leaving radiance contribution in addition to reducing the wavelength dependence of the plume strength, which is the difference between radiance emitted by plume and ground as expressed in Eq.7. The success of the atmospheric correction depends on the accurate characterization of the atmospheric state that is input into the radiative transfer model (RTM) e.g. MODTRAN (Berk et al., 2005). Independent atmospheric profiles of temperature, water vapor, and other gas

constituents (e.g., ozone) are input to the RTM to obtain the atmospheric transmittance, path radiance, and sky irradiance terms. Once the residual effects of the atmosphere have been removed from the observed radiance the surface properties can be obtained.

For multispectral data, where the bands are typically not strongly affected by the atmosphere the RTM approach works satisfactorily, but for hyperspectral data the RTM approach is more challenging when bands are situated in strong atmospheric absorption features and if output model data from the RTM are not accurately spectrally registered with the observed data, then

the solution may be unstable. This instability primarily arises because 1) methods used in the RTM to interpolate hyperspectral absorbances introduce error, 2) the sensor's spectral responses functions are not precisely defined, and 3) band-to-band registration issues result in model error. In these cases even small misregistrations between the observed and modeled data near strong absorption lines will amplify instead of reduce the effects of atmospheric attenuation, making correction of the radiance spectrum very difficult. An example of this issue is shown in Figure 4 where surface brightness temperature spectra are shown from HyTES

after atmospheric correction using the RTM approach with MODTRAN (gray line), and the ISAC approach (black line). With a successful atmospheric correction we expect a nearly constant temperature across all bands, which is achievable with the ISAC approach but not with MODTRAN below 8 μm and above 11.5 μm because of band-to-band misregistrations between HyTES data and MODTRAN in the presence of higher water vapor absorption regions.

For HyTES we use a modified version of the in-scene atmospheric correction (ISAC) approach initially developed for

the SEBASS airborne hyperspectral sensor (Young et al., 2002). The main advantage of the ISAC method is that atmospheric correction is accomplished using the hyperspectral data itself without the need for external atmospheric profiles or an RTM. In addition, the issue of spectral band misregistrations is eliminated. ISAC relies on finding graybodies in a given scene with emissivity close to 1 across all bands, $\epsilon_\lambda \sim 1$ (e.g., water, dense vegetation, ice, snow). Then the observed radiance in (2) can be written as a linear function with an independent variable, $B_\lambda(T_s)$, and with slope $\tau_\lambda^{atm}$ and y-intercept $L_\lambda^\downarrow$ as follows:

$$L_\lambda^{\text{off}} = \left[ B_\lambda(T_s)\tau_\lambda^{atm} + L_\lambda^\downarrow \right] \qquad (8)$$

Theoretically, the atmospheric parameters $\tau_\lambda^{atm}$ and $L_\lambda^\downarrow$ can then be found by simple linear regression by plotting $L_\lambda^{\text{off}}$ vs $B_\lambda(T_s)$ for all pixels on a scene for a given wavelength. We found that using the maximum brightness temperature *'most hits'* method as proposed by *Young et al.* (2002) resulted in pixels consisting of different types of soils in agricultural environments with emissivities <0.95 often being included in the fitting procedure. This was verified by comparing these pixels with high spatial resolution (100 m) emissivity information from the ASTER Global Emissivity Database (ASTER GED) (Hulley et al., 2015b).

Misclassification was usually worse over scenes with high temperatures, where bare soils exhibit near-blackbody like behavior and are confused with true graybodies such as dense vegetation. These non-graybody pixels violate the intrinsic assumptions of the ISAC method leading to errors in the fitting procedure.

To address this issue we developed a spectral variance approach in which the spectral variance in observed radiance was calculated for each pixel and only those pixels with low variance (e.g. a threshold set at less than 8 W/m^2) were assumed to be

graybody pixels suitable for use in the fitting procedure. Using this approach resulted in a very good match with graybodies classified according to the ASTER GED emissivities. The spectral variance approach is a good assumption for low altitude flights (1 km AGL) in which observed radiance is still representative of underlying surface spectral features, and also because the emissivity spectra of graybody surfaces such as vegetation, snow, ice, and water are pseudo-invariant in the 8-12 μm range.





A large fraction of the HyTES target sites including those over a few key methane hotspots (e.g. Kern River Oil Field) were flown over bare regions containing very few graybody pixels (e.g. vegetation, water) and an alternative ISAC approach had to be developed. In this approach, termed the 'ISAC-ASTER' method, emissivity information from the 5 ASTER GED TIR bands from 8-12 µm were used directly in the ISAC fitting procedure instead of relying on the blackbody assumption. ASTER GED

emissivities at 100 m spatial resolution were first geolocated and interpolated onto the HyTES scene and then a Principal component (PC) regression approach (Borbas et al., 2007) was used to extend the 5 ASTER band emissivities to the 256 HyTES bands from 7.4-12 µm (Hulley et al., 2015a).

### 4. Plume Detection Methodology

The problem of identifying plumes from trace gas species in hyperspectral data is based on a set of linear algebraic

expressions that are used to find signals in non-linear noisy (cluttered) background data (Funk et al., 2001; Theiler and Foy, 2006). The challenge is to condense a set of nonlinear results; radiative transfer through the atmosphere, and hyperspectral data, into a linear signal-in-noise problem. This approximation becomes easier with weaker plumes that are close to being linear in their effect on the observed signal. The problem can be further be simplified by transforming the radiance data to atmospherically compensated brightness temperatures. Several 'matched filter' formulations have been developed, each with a basic goal of generating a

weighting function based on a given specific target gas signature, and producing an image using the observed hyperspectral data in which the intensity of the image correlates with the presence of the desired signature assumed to be distinct from the background covariance. Example of transmittance spectra of various trace gases including $H_2O$, $CH_4$, $NO_2$, $H_2S$, $SO_2$, and $NH_3$ in different wavelength ranges in the thermal infrared and convolved to the HyTES spectral response functions are shown in Figure 5. The strongest $CH_4$ absorption feature at 7.68 µm has minimal overlap with strong water vapor absorption features on either side at 7.6

and 7.78 µm, allowing higher signal to noise detection during humid conditions. The strongest features of $H_2S$ and $SO_2$ are in the 8-9 µm range, while ammonia has distinct spectral features in the 10-11 µm window range in which $H_2O$ absorption is minimal. A key advantage of HyTES is its higher spectral resolution with respect to other airborne hyperspectral TIR sensors (see Table 1), which allows a clear separation between individual absorption features.

### 4.1 Clutter Matched Filter (CMF)

Starting with a datacube, $L$ of hyperspectral thermal infrared data, containing an image of $N$ columns by $n$ rows, where the columns denote the number of pixels in a given image, and n denotes the number of spectral channels. The goal is to find a wavelength-dependent spectral signature, $b$, which is assumed to be linearly superimposed on the background signal or clutter. This can be expressed by the following equation:

$$r = \propto b + c \qquad (9)$$

Where r is the total radiance and can be modeled as a linear combination of signal, $\propto b$, where $\propto$ is the strength of a plume

signature, $b$, and $c$ is a noise term that contains both sensor noise and scene clutter. The plume signature $b$ is usually expressed in terms of absorbance, and is typically extracted from the HITRAN database and convolved to the sensor's spectral response. Figure 6 shows an example of $CH_4$ and $H_2O$ absorbance spectra in the 7.5-8.1 µm range convolved to the HyTES spectral response and normalized from [0 1]. The scene clutter contains radiance contributions from the ground and atmosphere, and is defined as noise with cross-spectral correlations. These spectral cross-correlations can be written in terms of a covariance matrix, $K$:

$$K = \langle cc^T \rangle = \frac{1}{N} LL^T \qquad (10)$$



Given the covariance of the background clutter, $K$, we can then find the optimum filter vector, $q$, as follows:

$$q = \frac{K^{-1}b}{\sqrt{b^T K^{-1} b}} \qquad (11)$$

Where $q$ is normalized such that the variance, $q^T K q = 1$. This ensures that in the absence of the signal the matched filter image will have a variance of one. The final clutter matched filter (CMF) image, is calculated by applying $q$ to the original data cube of radiance:

$$CMF = q^T L \qquad (12)$$

In order to minimize the effects of striping and other noise in the data, the CMF is calculated in a column-wise fashion (along-track) for a given data swath. The CMF result for each column is then demeaned by subtracting the sample mean from each observation and dividing by the standard deviation using all pixels on the scene. This results in a mean CMF of zero and standard deviation of 1 for each column of data. The final CMF will produce an image in which the intensity correlates with the desired plume signature as defined by $b$. Values that are classified as outliers in the final CMF are strong evidence for the presence of the

desired signature, and their significance quantified by number of sigmas of the distribution, however this metric is only valid if the matched filter distribution is Gaussian (Funk et al., 2001).

We can further define a dimensionless quantity called the Signal Clutter Ratio (SCR), which is computed by applying the signal filter vector in (11) to the target plume signature, $b$

$$SCR = q^T b \qquad (13)$$

The SCR can be used as a metric for evaluating the strength of the desired target signal above background 'clutter', or the radiance

emitted by other targets in the field of view. Usually the optimally derived CMF in (12) will maximize the SCR values derived in (13). SCR values are normalized from [0 1] and values closer to 1 indicate higher confidence in the presence of the desired gas target pixels in the image data.

## 4.2 Plume Dilation Algorithm

The CMF detection algorithm for HyTES is optimized to detect only the strongest $CH_4$ sources using a 5-step process.

The algorithm is designed to minimize false positives while enhancing plume structure around the strongest sources using a plume dilation algorithm. This algorithm is used to provide qualitative information to help attribute emissions to specific source types and source locations. The CMF can also be tuned to detect more diffuse $CH_4$ enhancements that could be the result of advection further downwind from a specific source. For example ground surveys have shown that some of the highest concentrations are found downwind at significant distances (hundreds of meters) from the original source (Leifer, 2014). However lowering the CMF

threshold comes at the cost of increasing the likelihood of false positives in the final image.

Once a binary image of the strongest plume pixels generated from thresholding the CMF result, a dilation algorithm is used to enhance the structure and edges of the plume (Broadwater et al., 2008). The binary image is first dilated within a 2-by-2 pixel neighborhood and then multiplied by the original CMF detection image. This results in an image with modified CMF values in the neighborhood immediately around the original plume pixels. A slightly lower detection threshold is then applied to the new

detection image resulting in a binary image that is again dilated within a 2-by-2 pixel neighborhood. This process repeats until a minimum detection threshold is reached based on the initial threshold set. After each iteration, a contiguity test is applied that removes any pixels with less than 2 neighbors. The result is an adaptive plume growing algorithm that finds the gas plume edges immediately surrounding the strongest gas plume pixels, while simultaneously reducing any false positives and noise.

A number of different configurations and thresholds were tested, which resulted in a final set of steps that both optimized

the presence of the strongest gas plume pixels, while simultaneously reducing any false positives and noise. The result of the three





primary steps are demonstrated in Figure 7, which shows an example of a methane plume detected by HyTES over the Kern River Oil Field in the SJV on 5 February 2015.

### 5. Results of field testing

5       In this section we summarize results of field tests that evaluated the performance of the CMF plume detection and imaging capability for different gases, key emission sectors and a variety of surface conditions. This represents a small subset of a two year program spanning multiple seasons that ranged from test facilities in Wyoming, to oil and gas fields in Colorado and New Mexico to California's San Joaquin Valley to the Los Angeles basin.

### 5.1 Anthropogenic methane

      While HyTES has the ability to detect multiple trace gases, much of this work focused on improving understanding of
atmospheric methane given its high importance both for scientists and decision makers as a key climate-forcing greenhouse gas and ozone precursor. The atmospheric growth rate of methane and controlling emission sources remain highly uncertain at regional to global scales (Dlugokencky et al., 2009; Kirschke et al., 2013; Miller et al., 2014; Rigby et al., 2008). Future changes in surface temperatures and precipitation have the potential to dramatically alter natural methane fluxes from large Arctic reservoirs (Damm et al., 2010; Kort et al., 2012) and tropical wetlands (Dlugokencky et al., 2009), while transformational changes in anthropogenic
emissions from fossil fuel production threaten to further increase atmospheric methane abundance (Larsen et al., 2015). Examples of anthropogenic sources of methane include the natural gas and oil supply chains (production, storage, transmission, distribution, consumption), agricultural activities (enteric fermentation, manure management, rice cultivation), landfills, coal mining, stationary and mobile combustion, and wastewater treatment (Thomas and Zachariah, 2012). This work focuses on anthropogenic point source emitters rather than more diffuse area sources given that the former are both uncertain and more readily detectable with TIR
imaging spectroscopy.

      Detection of methane from infrared measurements is possible due to the absorption from strong rotational-vibrational transitions ($\nu$4) in the 7.3-8 μm range that have sufficient separation from the strong water vapor band centered at 6.3 μm (see Figure 6). Hyperspectral satellite sensors like the Infrared Atmospheric Sounding Interferometer (IASI) (Aires et al., 2002), the Tropospheric Emission Spectrometer (TES) (Beer, 2006), and the Atmospheric Infrared Sounder (AIRS) (Tobin et al., 2006) are
able to take advantage of these absorption characteristics of methane, however are limited by their coarse spatial resolutions (10 km or more) and insensitivity to near-surface concentrations due to sensor saturation issues. Airborne hyperspectral TIR sensors such as HyTES and others detailed in Table 1 have the imaging capability of detecting methane emission sources at the few-meter scale, allowing improved characterization of individual point sources towards better understanding of their distribution.

#### 5.1.1 Oil production example: Kern River Oil Field

30       HyTES flew a set of flightlines over four days covering the extent of the Kern River and Kern Front Oil Fields during June 2014 and February 2015. This is a relatively large (44 km$^2$) oil field in the greater Bakersfield area of California, densely populated with production wells, storage, processing and distribution infrastructure. Most of the production in this area relies on thermal enhanced oil recovery (EOR) technologies (e.g. steam flooding). This often results in mix of $CH_4$ gas and a high-temperature steam 'cloud' with high water vapor loading, which has the potential for confusing the matched filter for methane





detection resulting in false positives. Together with the complex terrain and often strong winds this offered a challenging test of the HyTES detection capability.

HyTES surveyed the Kern River oil field on 8 July 2014 with 9 flight lines (each 1 km wide by 10 km long), 10 flight lines on 5 February 2015, and an additional 10 flight lines on 8 February 2015 at an altitude of 1 km AGL with a pixel resolution

of ~2 m. The 2015 flight campaign was used to identify persistent sources, to refine the atmospheric correction and CMF visualization algorithms, and to identify priority targets for follow-up quantitative retrieval analysis with a more computationally intensive algorithm (Kuai et al., 2015). Using the CMF algorithm with a target spectrum of methane, multiple individual sources of methane were identified over the Kern River field. A number of these sources were persistent with detections in July 2014 and February 2015. Repeated detections over time provide confidence in the detection algorithm, especially when plume shapes and

trajectories correspond well with wind vector and speed observations from the same day. Examples of two of these persistent plume sources are illustrated in Figure 9 (a) and (b) for source A4 and (c) and (d) for source B1. Each panel shows the CMF for $CH_4$ overlayed on a surface temperature image derived from the HyTES longwave TIR data. Higher intensity CMF values in red/yellow are indicative of higher concentration of the target gas ($CH_4$). Spatial variations in the plume shapes are caused primarily from fluctuations in wind direction and speed, and also turbulence. The detected plumes all had high SCR values ranging from

0.75 - 0.85 and the shapes of all plumes were consistent with the wind direction derived from local meteorological measurements (Note the different wind direction and plume trajectories for source B1).

To illustrate the ability to distinguish methane from water associated with steam flooding in the Kern River field, Figure 10 shows an example of HyTES observed brightness temperature spectrum in the 7.5-8 µm range extracted from an on-plume and off-plume pixel for a plume detected over a well pad. The right image in Figure 10 shows the CMF overlay result with the on- and

off-plume pixels highlighted. The off-plume pixel was chosen to be similar in spectral shape and magnitude as the on-plume pixel, except without the evidence of methane absorption. Both spectra clearly show the strong water absorption feature in the 7.55-7.76 µm and 7.85-7.9 µm regions from ambient atmospheric water vapor loadings, while the distinctive $CH_4$ absorption feature between 7.65-7.7 µm is only seen for the on-plume pixel with a difference of ~10 K from the off-plume spectra. Figure 10 clearly shows a distinct separation between the $H_2O$ and $CH_4$ absorption features for the on-plume pixel due to the high spectral resolution of

HyTES data (18 nm spectral resolution).

### 5.1.2 Manure management example: Bakersfield dairies

Methane emissions associated with livestock represent the largest source of methane emissions in California; enteric fermentation contributes about 35%, and manure management about 30% of the total budget (EPA, 2011). In addition to methane, ammonia, hydrogen sulfide, and other oxygenated organic compounds are emitted from management of animal waste (manure).

At many dairies in the SJV, waste is flushed from animal houses into waste lagoons and storage ponds for storage and intermediate treatment (Ham and DeSutter, 2000; Liang et al., 2002; Ro et al., 2013).

HyTES conducted flights over dairy farms in the vicinity of Bakersfield during July 2014 and February 2015. Using the CMF method, HyTES identified methane source emissions from a number of different dairy farms in the southern Bakersfield dairy region that were concentrated primarily over anaerobic lagoons. Figure 11 shows an example of methane detected over a

dairy from a HyTES flight on 8 February 2015. A section of the HyTES swath (~2-km wide) is shown as grayscale temperature image with methane detected pixels overlayed in green. A distinct and localized methane source can be seen in the vicinity of a covered anaerobic lagoon in Figure 11, and the dispersion of the detected plume is consistent with wind measurements in the local area (from NNE at 0.5-3 m/s).





### 5.1.3 Controlled release experiment

On April 28, 2015, we worked with Pacific Gas and Electric to conduct a controlled release of natural gas from one of their pressure regulating stations near Bakersfield, California. Gas was released at 3 flux rates: 250, 500, and 1000 Standard Cubic Feet per Hour (SCFH), with a control accuracy of ~10%. The test lasted for about 3 hours around on solar noon, during which a total of 14 HyTES overpasses were conducted at a flight altitude of ~500 m. Ground measurements included a weather station and in-situ gas analyzers sampling methane mixing rations 1 meter above the release point and mobile transects of the downwind plume using an automobile. The goal of the experiment was to establish a minimum threshold of detection for the HyTES instrument based on a range of flux rates, and better understand the correlations between the CMF and concentration retrieval results.

Figure 12 shows an example of HyTES detected methane over the controlled release site (shown in photograph) at 19:38 UTC. In the image, higher intensity green pixels correspond to higher methane mole fractions beneath the HyTES aircraft. The brightest green pixels (red circle) indicate the approximate location of the release point, while lower intensity pixels can be seen advecting down the road in the southerly direction, which is consistent with the wind direction measured nearby at this time (2 m/s at 20°). In addition to the CMF, we also show results from the HyTES $CH_4$ quantitative retrieval algorithm (Kuai et al., 2015). The quantitative retrieval algorithm was developed and adapted from the algorithm used for retrieving trace gases from the Tropospheric Emission Spectrometer (TES) onboard the Aura Satellite. Using HyTES radiance spectra in the 7.5 to 9.2 μm range, the HyTES $CH_4$ quantitative algorithm has been used to retrieve methane partial column mole fractions with a total error of approximately 20% using uncertainties determined primarily from instrument noise and spectral interferences from air temperature, surface emissivity, and atmospheric water vapor (Kuai et al., 2015).

Table 2 shows details of the HyTES flight altitude, $CH_4$ flux released, wind speed, pixel size, and values of the maximum (dMax) and total accumulated (dTotal) values estimated from the CMF (unitless) and concentration retrieval (ppm) algorithms. The dMax value of the quantitative retrieval represents the maximum methane detection above background calculated for pixels in the immediate vicinity of the release point; $dMax = \max(CH_4) - BKG$, where $BKG$ is the average methane background retrieval located away from the plume in the same scene.. Similarly for the CMF result, the dMax represents the pixel with the maximum CMF value for detected plume pixels above the average background CMF value. Similarly the total values (dTotal) for the CMF and quantitative retrieval in Table 2 represent the sum of all detected plume pixels as identified by thresholding the CMF values; $dTotal = \sum_{i=1}^{n} CH_4(i) - BKG$, where $n$ is the total number of detected plume pixels.

Figure 13 shows scatterplots of dMax and dTotal $CH_4$ mole fraction (ppm) and CMF values (unitless) for the three different flux rates (250, 500, 1000 SCFH), with both quantities increasing with flux rate. The errorbars for the CMF were determined from the CMF variance across detected plume pixels, and for the quantitative retrieval were determined from the retrieval error analysis of various sources (e.g. air temperature, emissivity, water vapor). The quantitative retrieval and CMF results have high correlation (0.992 and 0.988) for both dMax and dTotal metrics, which gives confidence in using the more efficient CMF method to rapidly detect and attribute methane plume point sources when compared to the more rigorous and slower retrieval approach (<0.1 seconds/pixel for the CMF as opposed to ~12 seconds/pixel on average for the retrieval). These results also give confidence in using the CMF and retrieval approaches in a synergistic manner, for example the CMF approach could be used to first rapidly detect and identify locations of methane plumes from a large aerial survey, and using this information, selected plumes can be quantified in a more rigorous manner with full uncertainty statistics using the quantitative retrieval approach.





### 5.2 Multiple Chemical Species Detection

The following section demonstrates a few examples of the capability of HyTES to detect multiple chemical gas species. The ability to distinguish between different trace gas signatures within a single plume consisting of several contiguous pixels is a key advantage of TIR hyperspectral data. The different chemical species that will be shown include $NH_3$, $SO_2$, $H_2S$, $SO_2$, and $CH_4$
and their distinctive features in the infrared domain from 7.5 -12 micron are shown in Figure 5.

### 5.2.1 El Segundo Refinery and Power Plant, Los Angeles

HyTES surveyed a refinery and natural gas-fired plant in El Segundo, California on 5 July 2014. The purpose of this flight and other flights over industrial facilities in this region was to demonstrate the capability of detecting multiple chemical trace gas species simultaneously from different processes. This capability could be used in the future to efficiently monitor both regulated
and fugitive emission sources in industrial zones that are challenging to detect from the surface. Detection of fugitive emissions from airborne imagery can provide key information to identify the problem and enable mitigation, as well as improve inventories.

HyTES flew two lines over the El Segundo facility at an altitude of 1.1 km AGL (pixel resolution 2 m). The target absorption spectra for $SO_2$, $NO_2$, $NH_3$, $H_2S$, and $CH_4$ (Figure 5) were extracted from the HITRAN 2012 database and used simultaneously with the CMF method to observe any enhancements in the vicinity of the plant. Figure 14 shows the area covered
by a HyTES flightline over the El Segundo refinery and a gas-fired powerplant. Insets show HyTES imagery of the five detected trace gases highlighted in different colors and overlayed on retrieved grayscale surface temperature data. Three examples are indicated where two different chemical species were detected simultaneously within the same plume consisting of several contiguous pixels: $NH_3$ and $NO_2$ were detected over the refinery, while $NH_3$ and $H_2S$ were detected in two distinct plumes over the natural gas power plant, both highlighted in 14. A distinctive $CH_4$ plume was also detected in the southeastern region of the
refinery and a $SO_2$ plume was detected at the power plant. It is beyond the scope of this study to determine the controlling process for each of these sources; however, $NO_2$ and $SO_2$ emissions are often products of combustion, and $NH_3$ and $H_2S$ are known to be produced from post-combustion pollution control technologies used in natural gas-fired power plants. In situ mobile surveys have also shown elevated methane levels in this vicinity (Hopkins et al. in press). Successful detection of a variety of different chemical species at such fine scale gives confidence in being able to detect similar emissions at other combustion powerplants and refineries
in addition to detecting $SO_2$ from natural sources such as over volcanic regions (Realmuto et al., 1994).

### 6. Discussion

The results presented here demonstrate the strength of high spatial resolution TIR imaging spectroscopy for detecting localized sources for a variety of chemical trace gas species including $CH_4$, $NH_3$, $SO_2$, $H_2S$ and NO2. Through spectroscopic analysis of HyTES TIR imagery using a Clutter Matched Filter (CMF) approach, we were able to detect elevated concentrations
of these trace gases in spatial patterns that, given the winds, appeared to be consistent with emission plumes from point sources. In most cases we were able to infer the specific location of these sources down to spatial scales of a few meters using accurate geolocation information provided with the HyTES data.

Atmospheric methane was detected over a wide variety of different sources including fugitive emissions from oil and gas fields, landfills, and dairies. From the 2014 and 2015 HyTES data campaigns, more than 100 individual point sources of methane
were characterized in the Kern River and Elk Hills oil and gas fields in the SJV, with most emissions originating from large infrastructure such as storage and processing facilities, and distribution pipes, rather than active well heads.



CMF plume imagery are useful for rapidly identifying the location of large and persistent point source emissions, including attribution of source types. This information has been used to focus subsequent analysis with more computationally intensive, quantitative retrieval algorithms (Kuai et al., 2015). 'Quicklook' CMF images can be generated on-demand for any specific target gas within a few hours of the observation time, although not part of routine HyTES processing, to assist with rapid

deployment of ground teams to measure in situ concentrations of the identified plumes using various instruments such as open-path in situ gas analyzers and thermal infrared cameras.

**7. Conclusion**

This study demonstrates the capability of the HyTES to detect and characterize atmospheric plumes of multiple trace gas species ($CH_4$, $H_2S$, $NH_3$, $NO_2$, and $SO_2$) for individual emission sources at high spatial resolution over larger areas (100's of km²)

under representative field conditions. HyTES produces wide swath Thermal Infrared (TIR) images at high spectral (256 bands from 7.5- 12 μm) and spatial resolution (~2 m at 1 km altitude), and presents a major advance in airborne TIR hyperspectral remote sensing measurements. HyTES can characterize the spatial extent and identify the specific source for individual gas plumes for moderate to strong emitters. Of particular interest is the characterization of methane point sources that remain highly uncertain.

Three HyTES science campaigns during the summer of 2014 and winter/spring of 2015 targeted a variety of trace gas

sources such as oil fields, gas pipelines, landfills, and dairies in the state of California. Using a hybrid Clutter Matched Filter (CMF) technique and plume dilation algorithm, HyTES successfully detected more than 100 discrete and persistent methane sources over the oil and dairy farms in the San Joaquin Valley (SJV), California. Spatial patterns of methane plumes detected by HyTES were consistent with coincident in-situ methane profile and wind measurements at the surface and from other aircraft. In addition to the HyTES plume detection/attribution capability, a HyTES methane concentration retrieval algorithm was developed

and adapted from the algorithm used for retrieving trace gases from the TES instrument onboard the Aura Satellite.

A controlled release experiment of methane gas in the Bakersfield region demonstrated that HyTES could detect methane fluxes as small as 250 Square Cubic Feet/Hr at 500 m flight altitude with ~ 2 m/s winds. The controlled release results also showed high correlation between the CMF and concentration retrieval results, which gives confidence in using these two approaches in a synergistic manner. For example the CMF approach could be used to first rapidly detect and identify locations of methane plumes

from broad aerial surveys, and then guide focused application of the full methane algorithm to generate quantitative estimates in a more rigorous manner with full set of uncertainty statistics to help address key science questions.

The quantitative retrieval capability combined with high resolution wind data will be used in the future to support emission flux estimation of methane point sources. The high spatial resolution imaging capability of HyTES for methane and other trace gas plumes will help fill an important niche in tiered observing strategies by complementing the larger coverage but coarser spatial

resolution offered by satellite methane observations and high measurement accuracy of mobile surface in-situ observations. Collectively, these measurement systems offer new tools for improving scientific understanding and decision making associated with methane emission sources.

**Acknowledgements.** The research described in this paper was carried out at the Jet Propulsion Laboratory, California Institute of

Technology, under contract with the National Aeronautics and Space Administration. Many thanks to Francois Rongere from Pacific Gas and Electric's R&D and Innovation division for their support for the controlled release test.



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



Table 1. Instrument characteristics of current airborne hyperspectral thermal infrared systems.

| Instrument | First Deployed | Bands | Spectral Range (µm) | Spectral Resolution (nm) | IFOV (mrad) | Max Scan (°) | Pixels X-track | NEDT* (K) | Detector |
|---|---|---|---|---|---|---|---|---|---|
| AisaOWL | 2014 | 96 | 7.7-12.3 | 100 | 1.10 | ±24 | 384 | 25** | MCT |
| HyTES | 2013 | 256 | 7.5-12 | 18 | 1.70 | ±50 | 512 | 0.20 | QWIP |
| MAGI | 2011 | 32 | 7.1-12.7 | 175 | 0.53 | ±42 | 2800 | 0.10 | HgCdTe |
| SIELETERS | 2011 | 38 | 8-11.5 | 80 | 0.25 | ±7 | - | 0.15 | HgCdTe |
| MAKO | 2010 | 128 | 7.45-13.5 | 47 | 0.55 | ±45 | 400-2750 | 0.05 | HgCdTe |
| SEBASS | 1995 | 128 | 7.6-13.5 | 46 | 1.10 | ±7.3 | 128 | 0.05 | Cooled Helium prism |

*NEDT = Noise Equivalent Differential Temperature (K)

5    ** NESR = Noise Equivalent Spectral Radiance (mW/m2/sr/µm)

*** IFOV = Instantaneous Field of View



Table 2. Results from a controlled release experiment on April 28, 2015 where natural gas was released from a ~ 2 m high stack at a pressure regulating station near Bakersfield, California. Aircraft altitude, fluxes (SCFH = Standard Cubic Feet per Hour), wind speed, pixel size, and Maximum (dMax) and total values (dTotal) of $CH_4$ calculated from the Clutter Matched Filter (CMF) and $CH_4$ retrieved concentration values (ppm) are shown, where dMax = [max($CH_4$) - BKG], and dTotal = [Sum($CH_4$) - BKG], where

BKG = average $CH_4$ background value of pixels in which no  plume was detected.

| | | | | CMF Results | | Retrieval Results | |
|---|---|---|---|---|---|---|---|
| Altitude (m) | Fluxes (SCFH) | Wind Speed (m/s) | Pixel Size (m) | dMax (% error) | dTotal | dMax (% error) (ppm) | dTotal (ppm) |
| 500 | 1000 | 1.96 | 0.782 | 0.55 (4.5) | 158.15 | 0.98 (12) | 6.24 |
| 500 | 500 | 2.30 | 0.754 | 0.49 (2.8) | 133.39 | 0.93 (21) | 3.64 |
| 500 | 250 | 1.94 | 0.785 | 0.27 (6.1) | 66.56 | 0.5 (18) | 1.29 |



**Figure Captions**

Figure 1. (a) HyTES design and optical layout, (b) Twin Otter aircraft, (c) HyTES installation in aircraft, and (d), optical layout highlighting ray trace through the Dyson spectrometer and objective lens elements.

Figure 2. HyTES data hypercube over Death Valley, California. Radiances in the vertical slice have been atmospherically corrected for the atmospheric transmission and path radiance.

Figure 3. Illustration depicting various components of thermal infrared radiative transfer with a gaseous plume, where: $L_\lambda^{\text{surf}}$ =

Earth-emitted radiance, $L_\lambda^p$ = plume thermal emission term, $L_\lambda^\uparrow$ = atmospheric path radiance, $\rho_\lambda L_\lambda^\downarrow$ = reflected downwelling radiance, $\tau_\lambda^{atm}$ = atmospheric transmittance, $\tau_\lambda^p$ = plume transmittance, $\theta$ = observation angle, and P(*) = Pressure level.

Figure 4. An example of surface brightness temperature spectra from HyTES after atmospheric correction using the RTM approach with MODTRAN (gray line), and the ISAC approach (black line). With a successful atmospheric correction we expect a nearly

constant temperature across all bands, which is achievable with the ISAC approach but not with MODTRAN below 8 μm and above 11.5 μm because of misregistrations between HyTES data and MODTRAN.

Figure 5. Transmittance spectra of $H_2O$, $CH_4$, $NO_2$, $H_2S$, $SO_2$, and $NH_3$ in different wavelength ranges in the thermal infrared. Transmittances were generated with MODTRAN 5.2 simulations using a standard atmosphere and sensor altitude of 3 km, and

convolved to the HyTES spectral response functions (256 bands from 7.4-12 μm).

Figure 6. Absorption spectra of $H_2O$ and $CH_4$ extracted from the HITRAN 2012 database and convolved to the HyTES spectral response functions in the 7.5-8.1 μm range. The square symbols represent the HyTES bands in this range.

Figure 7. An example of the 3-step plume detection and enhancement algorithm for two methane plumes detected over the Kern River Oil Field (top panels) and Four Corners (bottom panels), (a.1,2) original CMF normalized from [0 1] where brightest pixels are associated with the presence of the target gas plume, (b.1, 2) the CMF is thresholded using an Inter-Quartile Range with weight set to 2.5, and (c.1, 2) final plume image after a plume dilation algorithm is implemented (see text for details).

Figure 8. Higher-level plume visualization images showing a color CMF overlay on grayscale surface temperature image on a geospatial map. Higher values of the CMF (red) correlate higher with the intensity of the desired plume signature (in this case methane), and (inset) a Google Earth KMZ file containing a CMF overlay in green on a surface temperature image. This example shows the surface temperature imaged faded out to reveal only the green plume pixels overlayed on the visible Google Earth image.

Figure 9. Examples of persistent methane plumes detected by HyTES over oil condensate storage tanks in the Kern Front and Kern River oil fields near Bakersfield, California. Sources A4 in panels (a) and (b), and B1 in panels (c) and (d) were detected during July 2014 and February 2015. Plume enhancements are shown in color using the CMF method overlaid on a surface temperature image.




Figure 10. HyTES brightness temperature spectra (left) from 7.5 - 8 μm for an on-plume and off-plume pixel indicated in the CMF-temperature overlay (right) for a plume over the Kern River Oil Field in July 2014. The presence of $H_2O$ features in both the on-plume and off-plume pixels but only $CH_4$ in the on-plume pixel indicates the latter detection is not a false positive.

Figure 11. HyTES detected methane plumes over a dairy farm in the San Joaquin Valley, California during February 2015 displayed in Google Earth with the methane plume in green overlayed on HyTES grayscale surface temperature retrieval. The dispersion of the detected plume is consistent with wind measurements in the local area (from NNE at 0.4 m/s with gusts to 2.8 m/s).

Figure 12. Example of HyTES detected methane (green) with overlay on grayscale surface temperature (left) at the natural gas
controlled release site (inset photograph). Detected methane in the HyTES image is displayed in green with the higher intensity color corresponding with highest concentration of methane at the release point circled in red.

Figure 13. Scatterplots of maximum and total $CH_4$ concentration (ppm) and CMF values (unitless) for three different flux rates (250, 500, 1000 Standard Cubic Feet per Hour) at the controlled release site near Bakersfield, CA on 28 April 2015. The max values (left) represent the highest concentration/CMF values in the vicinity of the release point above background values, while
the total value (right) represents the accumulated sum of quantitative/CMF values over the detected plume pixels determined from thresholding the CMF result. The quantitative retrieval and CMF results have high correlation (0.992 and 0.988), which gives confidence in using the more efficient CMF method to rapidly detect and attribute methane plume point sources when compared to the more rigorous and slower retrieval approach (<0.1 seconds/pixel for the CMF as opposed to 12 seconds/pixel on average for the retrieval).

Figure 14. A HyTES Multi-species gas detection example showing a Google Earth image (center) of the area covered by a HyTES flightline over a refinery (magenta outline) and a natural gas powerplant (yellow outline) near El Segundo, CA. The insets show HyTES imagery of five detected trace gases ($CH_4$, $NO_2$, $NH_3$, $H_2S$, and $SO_2$) highlighted in different colors and overlayed on retrieved surface temperature data in grayscale. Three examples are indicated where two different gases were detected
simultaneously within the same plume consisting of several contiguous pixels; $NH_3$ and $NO_2$ were detected over the refinery at the location **a.1**,/**a.2**, while at the natural gas powerplant, $NH_3$ and $H_2S$ were detected at location **b.1**/**b.2**, and **c.1**/**c.2** respectively. Small plumes of $SO_2$ (blue) can also clearly be seen being emitted from areas of the power plant (inset photograph). A distinctive $CH_4$ plume was detected in the southeastern region of the refinery.





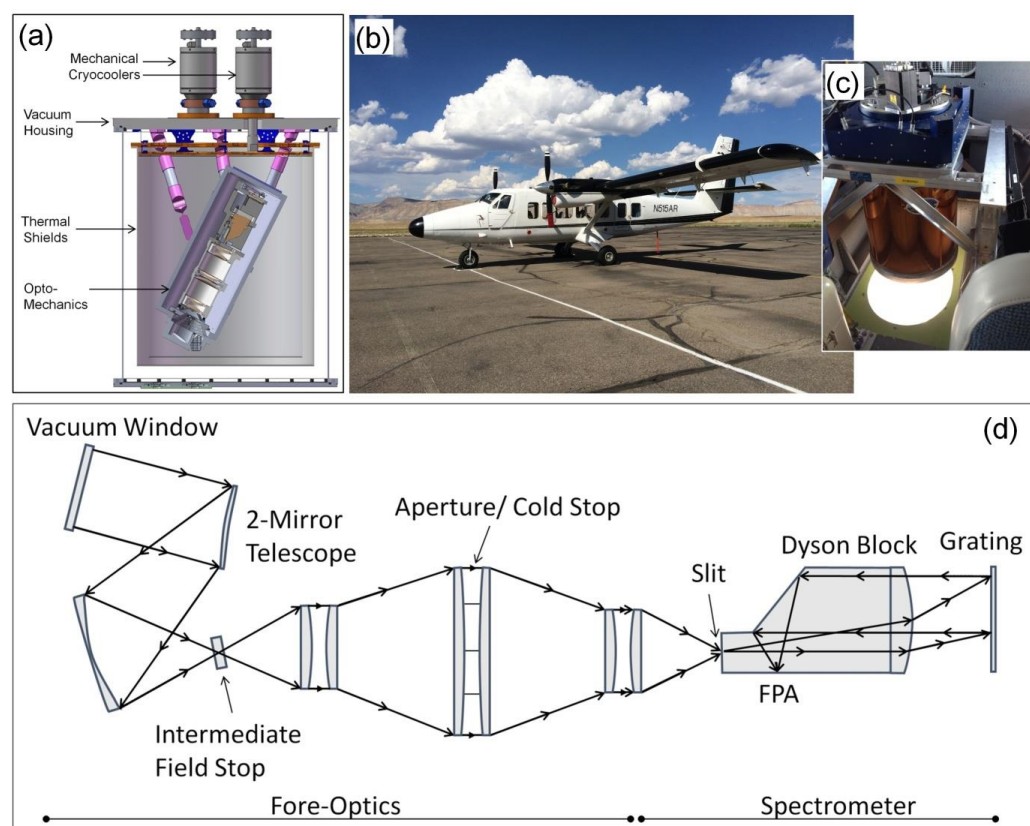

Figure 1.





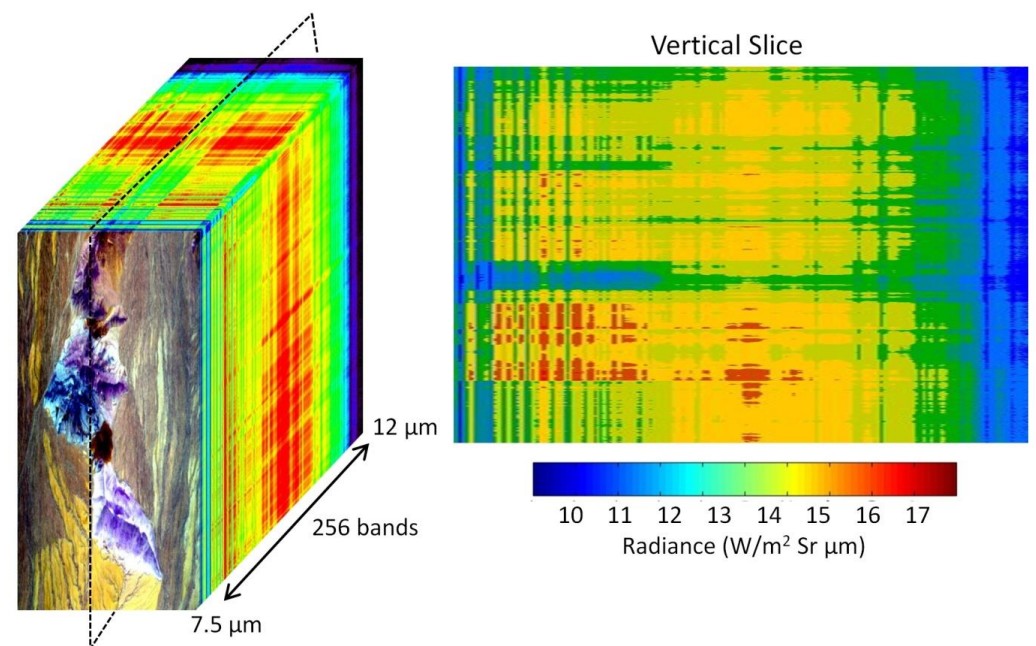

Figure 2.



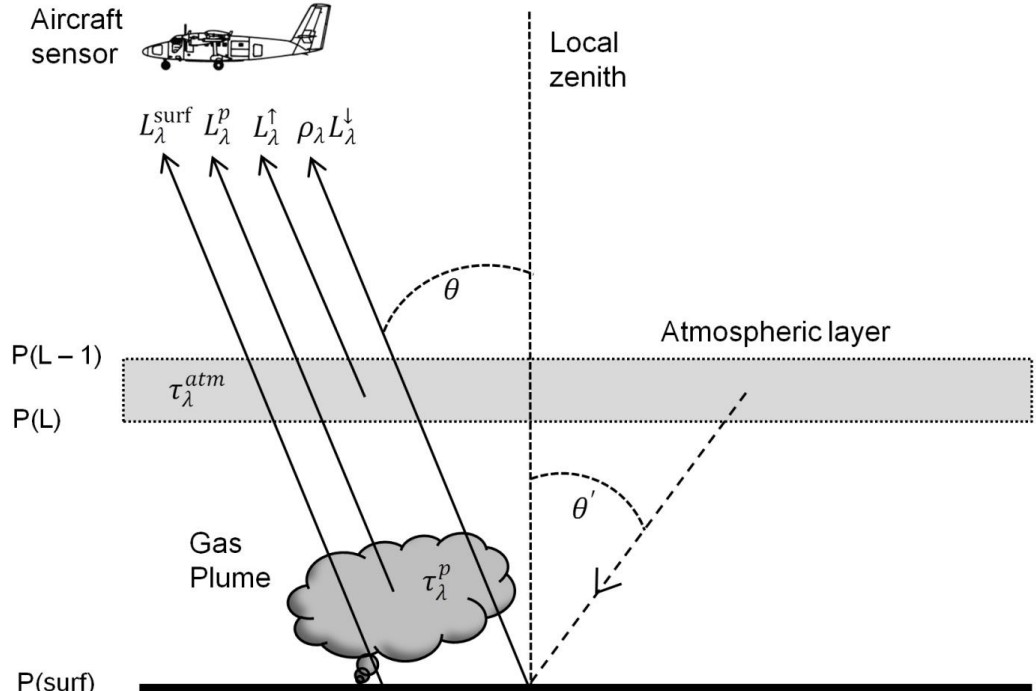

Figure 3.





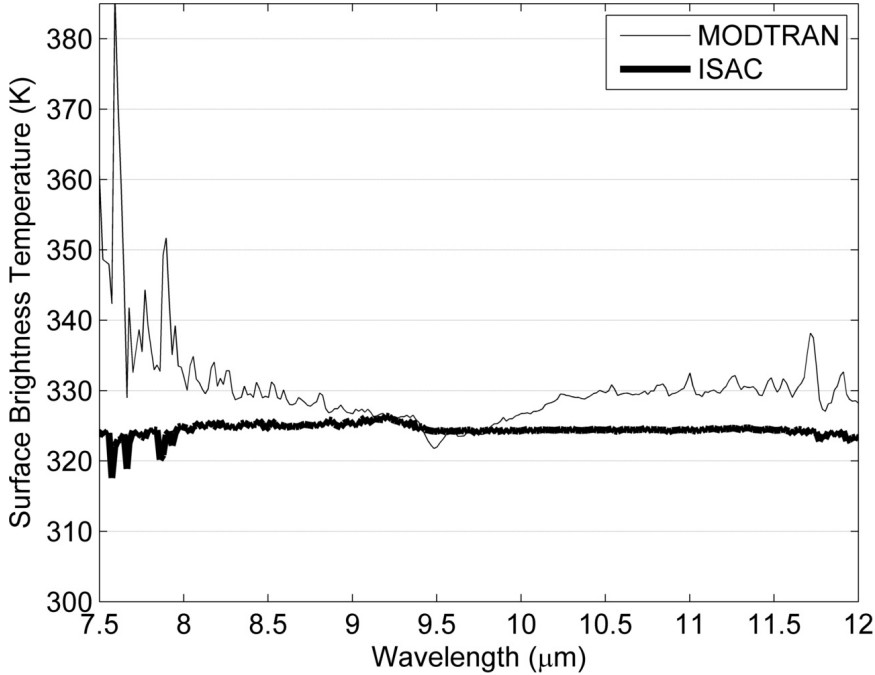

Figure 4.




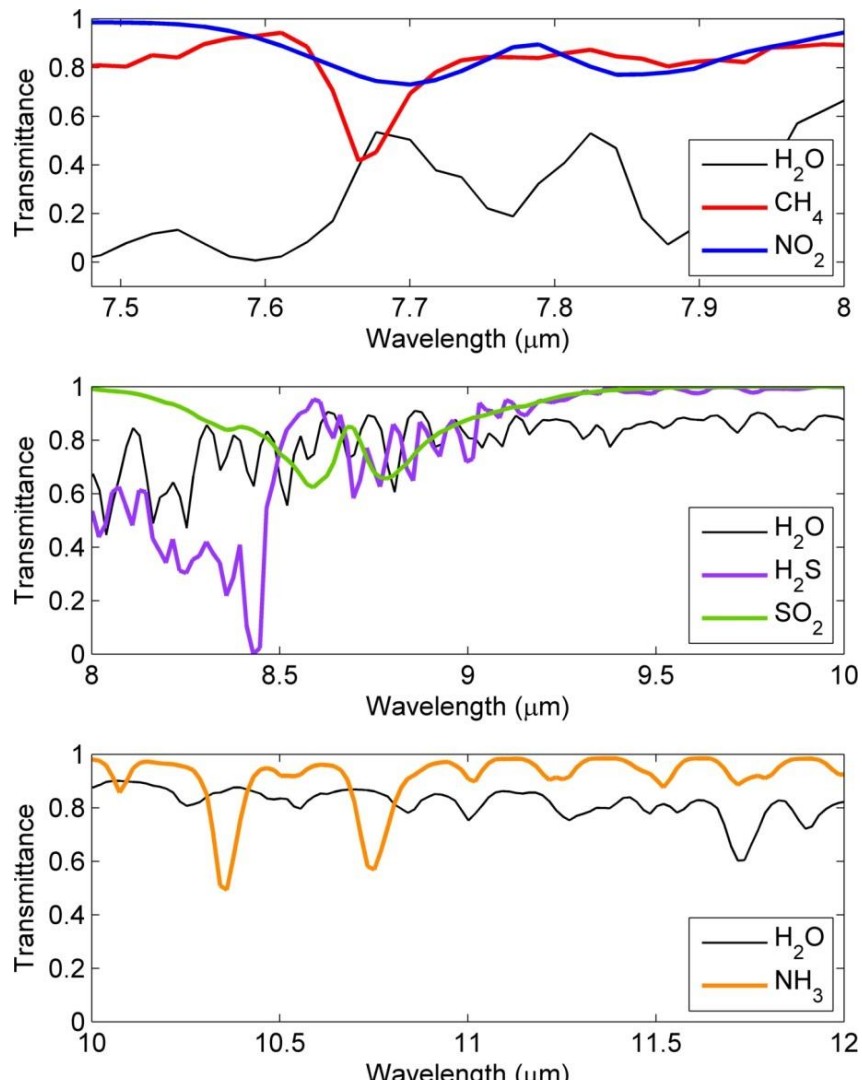

Figure 5.





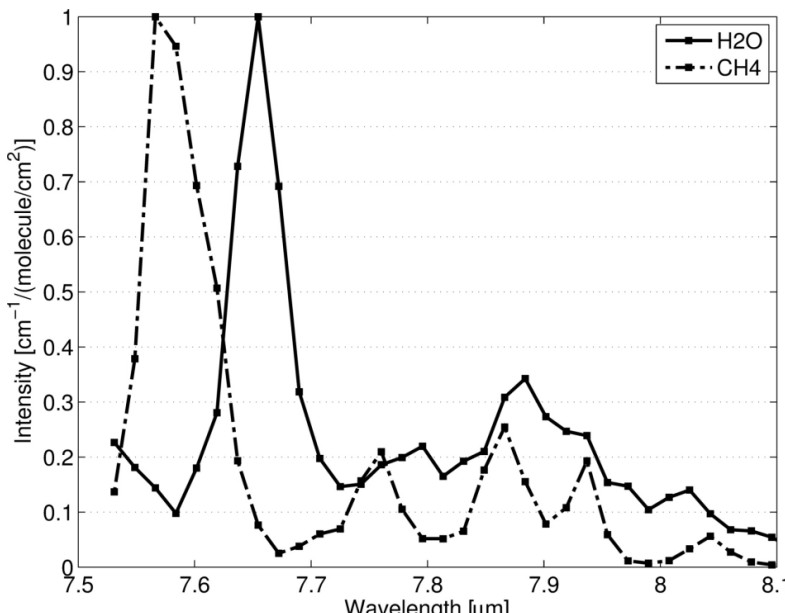

Figure 6.





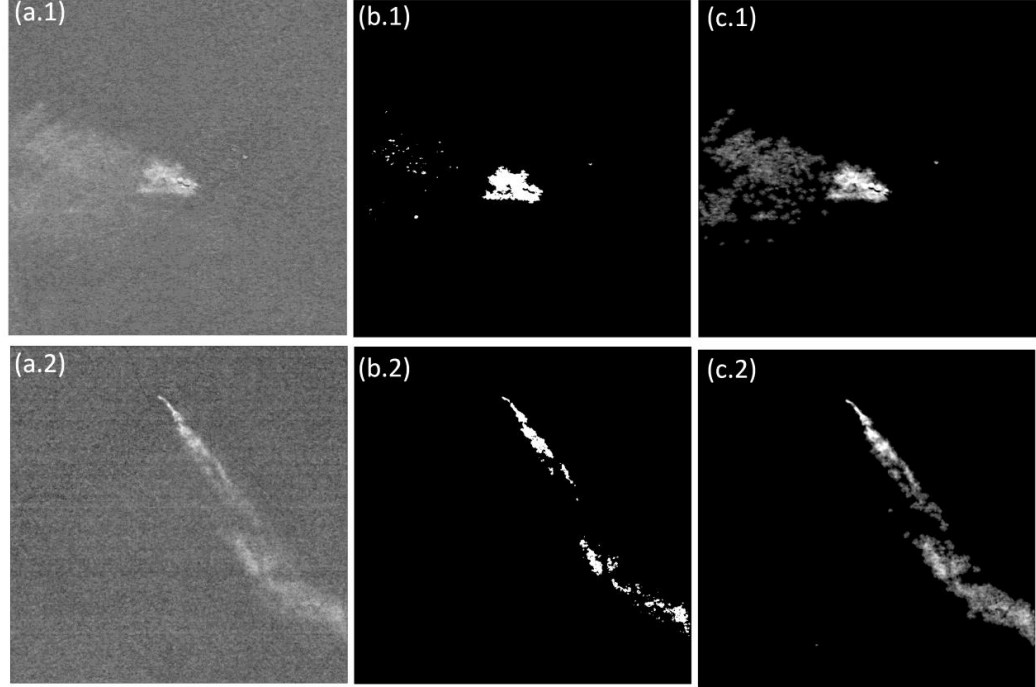

Figure 7.





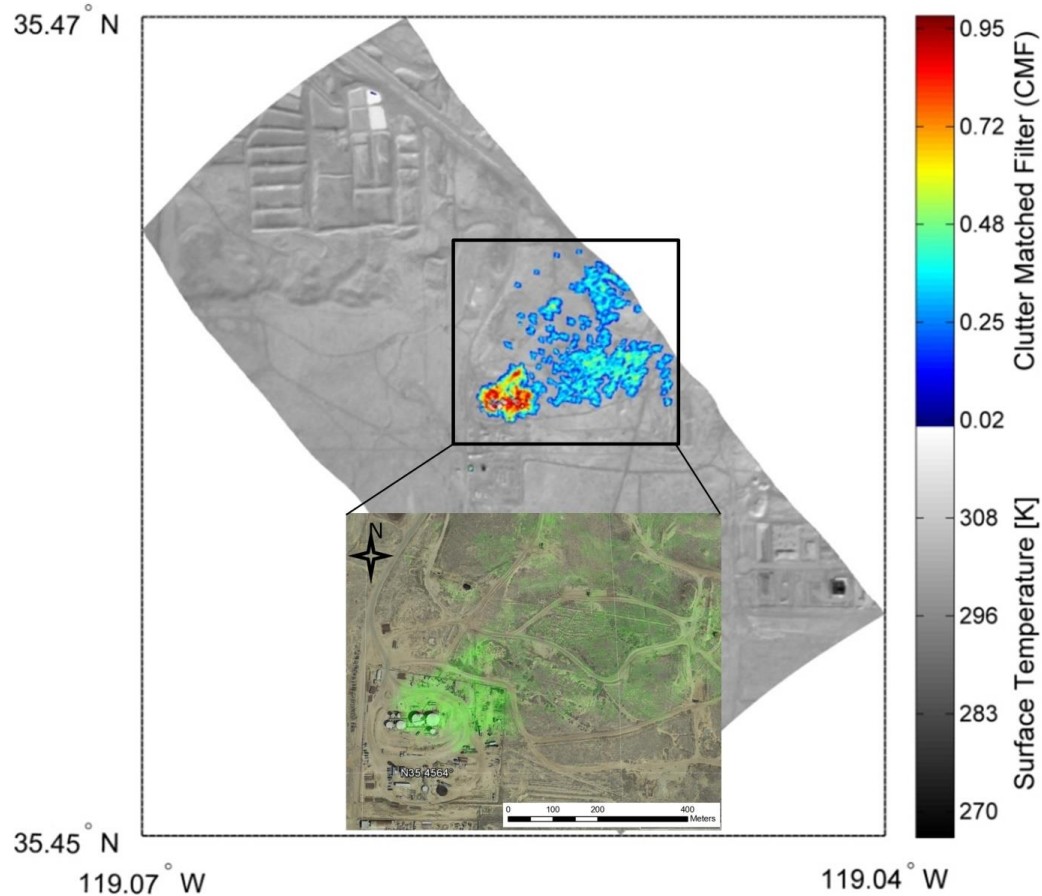

Figure 8.

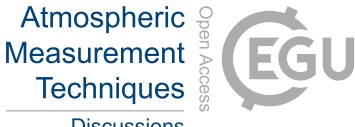



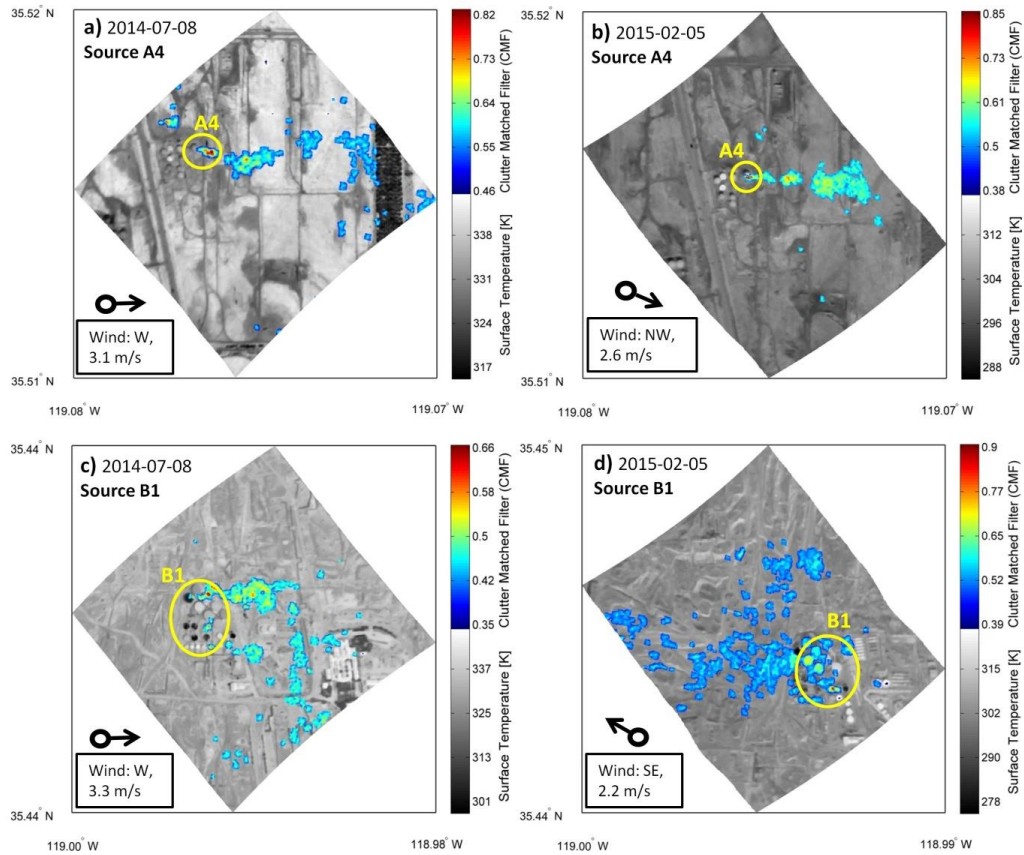

Figure 9.




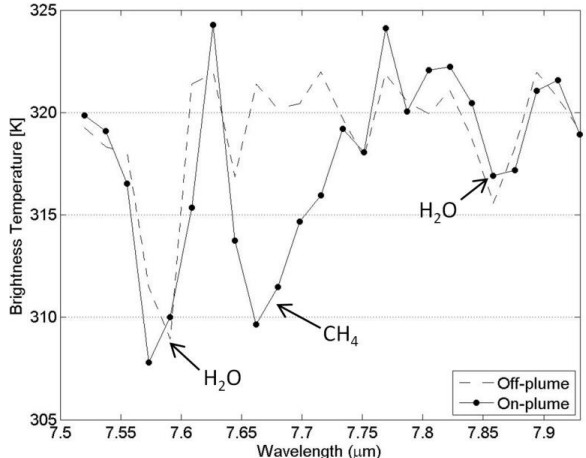
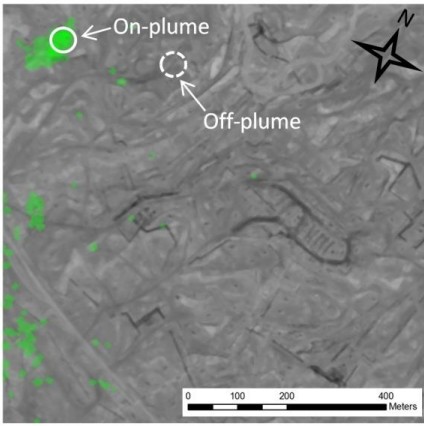

Figure 10.

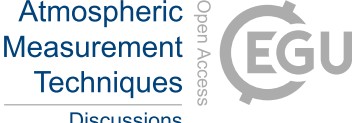



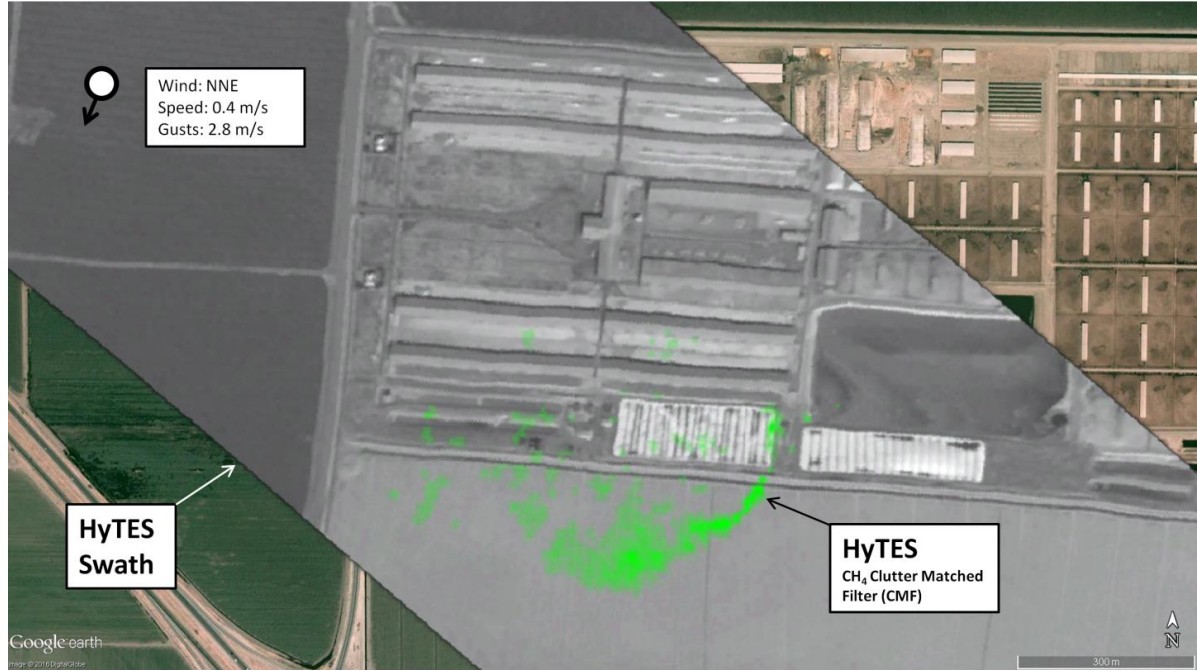

Figure 11.




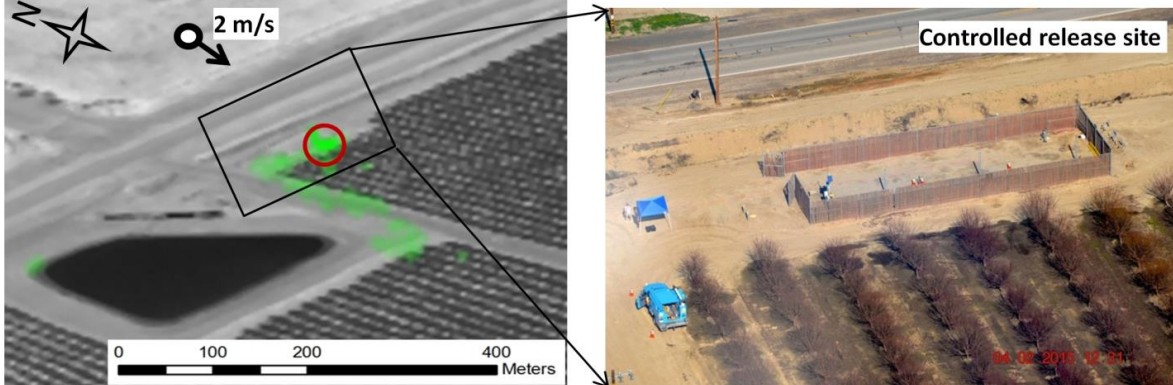

Figure 12.





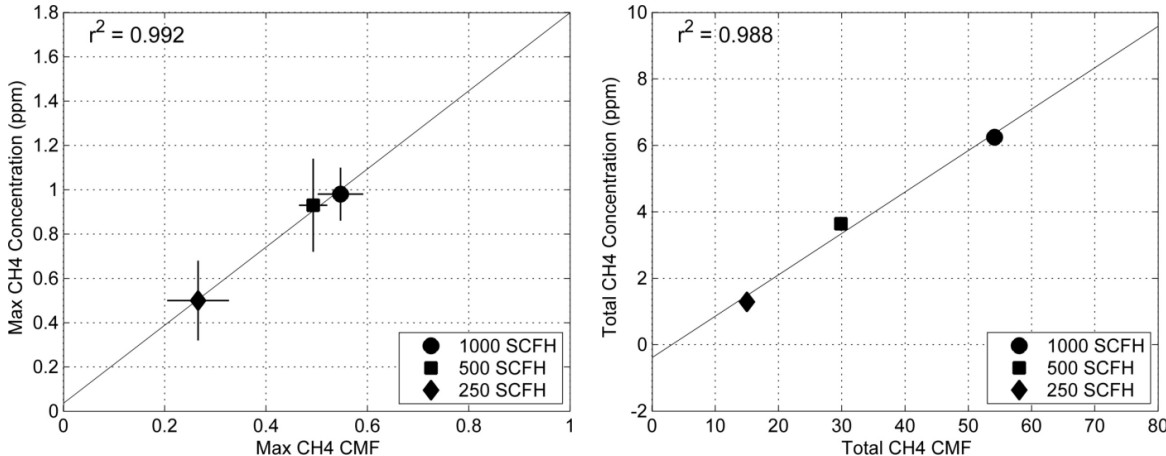

Figure 13.





Figure 14.

