# Peer review of "High spatial resolution imaging of methane and other trace gases with the airborne Hyperspectral Thermal Emission Spectrometer (HyTES)"

_Atmospheric Measurement Techniques, 2016_

## Referee Comment (RC1) · Anonymous Referee #1 · 29 Feb 2016

General Comments

Overall this manuscript presents some intriguing measurements of fugitive gas emissions using airborne TIR imaging spectrometry and makes a sufficient contribution to the growing body of literature on this topic to merit consideration by AMT.

There are a number of typographical items such as surplus prepositions that are easily rectified with the help of a grammar checking tool; these are not listed in this review and are left to the authors to fix. Beyond this minor issue, the manuscript contains a number of material deficiencies that should be addressed before acceptance by AMT.

Specific Comments

P2, L3-4: Seawater and vegetation do have low albedo in the SWIR, but both also tend to exhibit low thermal contrast with respect to the overlying airmass. So the advantage claimed for TIR over SWIR is not as "distinct" as it could be.

P2, L6-7: Methane hydrates decompose at the ocean surface. The authors presumably mean "ocean floor."

P2, L9-10: "Another key advantage of TIR hyperspectral data is the ability to distinguish between different trace gas signatures within a single plume."

This cannot be claimed as a particular advantage of TIR, since spectral unmixing is a technique that has been applied across several spectral bands for decades.

P2, L17-18: The authors class MAGI-L as an airborne sensor, yet the reference cited makes clear that it is a space concept, one moreover that has not flown. The authors clearly have the airborne MAGI in mind, for which a more appropriate reference would be doi:10.1139/TGRS.2015.2422817.

P2, L19-20: "…HyTES has the highest number of spectral bands (256) which improves the detection sensitivity of trace atmospheric gas constituents…"

This statement is too simplistic. Detection sensitivity could be enhanced by higher spectral resolution for substances that have narrow spectral diagnostic features (compared with the instrumental spectral resolution). This will be true for some gases, but by no means all.

In addition, when determining detection sensitivity the spectral resolution must be traded against the received radiance per channel. Thus the 256 channels of HyTES result in correspondingly reduced radiometric flux in each spectral channel. This is an important consideration for HyTES in particular, since its QWIP focal plane has a comparatively low quantum efficiency. The statement regarding detection sensitivity should therefore be discussed in greater depth.

P2, L36: "…provide in situ measurements to validate those results." Explain how this

was accomplished.

P3, L8-10: "…relatively low power requirements" with two cryocoolers? Relative to what?

P3, L13: "A single sensor calibration is used for an entire field campaign…"

TIR sensors are prone to calibration drift and a single calibration is never regarded as valid for even a single flight, let alone an entire campaign. The temperature of the Dewar window alone will change with altitude in the unpressurized airplane cabin. The authors' procedure calls into question the accuracy of their data.

P4, L19: TIR gas detection does not necessarily require a "strong" thermal contrast, but the contrast does have to be finite.

P5, L28-29: Can 100 m really be regarded as "high spatial resolution" in the context of this report?

P7, L5-6: Clarify what is meant by "column" in this context.

P7, L13-14: SCR is a dimensionless quantity, which does not comport with the RHS of Eq. (13). Eq. (6) of the Funk et al. (2001) reference provides the correct form of the SCR. This has significant ramifications if Eq. (13) really was used to assess signal strength as the authors state.

P11, L3-4: "The ability to distinguish between different trace gas signatures within a single plume is a key advantage of TIR hyperspectral data."

As already stated above, this is NOT unique to TIR and therefore cannot be claimed as a specific advantage.

P15: There are several problems with Table 1:

1. In the header, IFOV should be accompanied by 3 asterisks to link it to the legend at bottom.

2. For AisaOWL use HgCdTe in place of MCT, as has been done in the remainder of the table.

3. Both MAKO and SEBASS use Si:As blocked impurity band detectors (and incidentally, "cooled helium prism" refers, albeit in mangled fashion, to a dispersion element, rather than a detector).

4. The scan angles given for HyTES and SEBASS are approximately twice the actual values.

An overarching question about this table concerns what the criteria for inclusion were. The SIELETERS group provide a much more comprehensive tabulation of TIR sensors in a paper not cited here (see doi:10.1364/OE.23.016164). The authors' table would benefit from significant revision.

P17, L18-20: Caption and accompanying figure (P23): At least some of the transmittances shown in Figure 5 do not correspond to the standard atmosphere that the authors claim to have used (in particular, total extinction of the 8.4-micron H2S feature as shown is not credible – see also remarks for Figure 14 on P32, below). It would be instructive to provide the MODTRAN derived column densities for these spectra.

P17, L30-33: Caption and accompanying figure (P26): The description of the inset needs rewording. It's said to be a CMF image overlain onto surface temperature, yet the underlying image is actually from Google Earth. The last sentence of this caption only adds to the confusion, but maybe it's moot since the figure is neither cited nor explained in the text and can thus be simply deleted.

P32: Figure 14: The peak absorption coefficient for H2S is almost 3 orders of magnitude smaller than ammonia in the TIR band and is unlikely to be detected by this sensor unless its abundance is at potentially lethal levels (a reportable episode at the very least). Has a concentration estimate been computed for this observation?

Technical Corrections

P2, L22: The HyTES spectral range is everywhere else given as 7.5-12 microns, except here.

P5, L15: ISAC is invoked before being defined (P5, L19).

P5, L24-25: The "downwelling" radiances should actually be upwelling.

P10, L3-4: It's OK to use SCFH because it's an industry standard, but these values should also be accompanied by their equivalents in SI units.

P10, L13: I spent some time searching for quantitative results in Fig. 12 before realizing that the authors must have meant Fig. 13. Confusing.

P11, L4: SO2 is listed twice.

P11, L19: Insert "Figure" before "14."

P12, L22: "Square Cubic Feet/Hr." should of course be "Standard Cubic Feet/Hr." It's already defined above, so why not just use SCFH? In any case, as mentioned above, these values should be accompanied by their equivalents in SI units.

P13, L6-16, L26-27, L33-38, L48-50, L54-56; P14, L10-12, L15-16, L53-54: Citations incomplete.

P16: Table 2: Provide fluxes in SI units as well as SCFH.

P24: Figure 6: Y-axis should actually be "Normalized Absorption."

P31: Figure 13: Provide flow rates in SI units as well as SCFH.

---

## Author Comment (AC1) · 18 Mar 2016

**Response to Anonymous Referee #1**

P2, L3-4: Seawater and vegetation do have low albedo in the SWIR, but both also tend to exhibit low thermal contrast with respect to the overlying airmass. So the advantage claimed for TIR over SWIR is not as "distinct" as it could be.

**Yes we agree, which is why the sentence begun with *"For example, given sufficient thermal contrast between the plume and the surface..."*. However we agree that *'distinct'* may be too presumptuous, and have rephrased the statement as follows:**

*"For example, given sufficient thermal contrast between the plume and the surface, TIR data should on average have higher sensitivity to methane detection than SWIR data over low albedo surfaces such as seawater and dark vegetation, and particularly at higher latitudes where reduced reflective solar insolation make it a challenge for current SWIR instrument capabilities. This is because thermal contrast can change rapidly with local atmospheric conditions over much shorter time scales than the underlying reflective surface features such of water and dark vegetation of which the SWIR instruments are responsive to."*

P2, L6-7: Methane hydrates decompose at the ocean surface. The authors presumably mean "ocean floor."

**Actually ocean surveys have found CH4 to be supersaturated in surface waters of the Arctic far removed from continental shelves and attributed this observation to aerobic CH4 production (e.g. see Kort et al. 2012, Damm et al. 2010, and Damm et al. 2011).**

P2, L9-10: "Another key advantage of TIR hyperspectral data is the ability to distinguish between different trace gas signatures within a single plume." This cannot be claimed as a particular advantage of TIR, since spectral unmixing is a technique that has been applied across several spectral bands for decades.

**The reviewer is correct that spectral unmixing has been successfully applied across all spectral bands in both VSWIR and TIR, but mostly for geologic and compositional surface studies. In the context of trace gas detection, to our knowledge VSWIR measurements such as from AVIRIS-NG have the capability to detect and unmix signatures of greenhouse gases (CH4, CO2, and H2O) from a single observation, however, they do not have the ability to detect criteria pollutants such as H2S, NH3, NOx, and SO2, let alone unmix their signals from a single plume (personal communication, Andrew Thorpe and David Thompson, JPL). HyTES on the other hand has the ability to detect both greenhouse gases and criteria pollutants and unmix their signals from a single overpass. We clarified this capability in the text:**

*" Another key advantage of TIR hyperspectral data is the ability to distinguish between both greenhouse gases (e.g. $CH_4$) and criteria pollutants (e.g. $H_2S$, $NH_3$, $NO_2$, and $SO_2$) within a single plume - a capability that will be demonstrated in this work."*

P2, L17-18: The authors class MAGI-L as an airborne sensor, yet the reference cited makes clear that it is a space concept, one moreover that has not flown. The authors clearly have the airborne MAGI in mind, for which a more appropriate reference would be doi:10.1139/TGRS.2015.2422817.

**Thank you for the suggestion; this was a careless oversight on our part, we have added the suggested reference that describes the MAGI airborne instrument:**

**Hall JL, Boucher RH, Buckland KN, Gutierrez DJ, Hackwell JA, Johnson BR, Keim ER, Moreno NM, Ramsey MS, Sivjee MG, Tratt DM, Warren DW, Young SJ. MAGI: A New High-Performance Airborne Thermal-Infrared Imaging Spectrometer for Earth Science Applications *Ieee Transactions On Geoscience and Remote Sensing*.**
**DOI: 10.1109/TGRS.2015.2422817**

P2, L19-20: ": : :HyTES has the highest number of spectral bands (256) which improves the detection sensitivity of trace atmospheric gas constituents: : :"

This statement is too simplistic. Detection sensitivity could be enhanced by higher spectral resolution for substances that have narrow spectral diagnostic features (compared with the instrumental spectral resolution). This will be true for some gases, but by no means all.

**It is beyond the scope of this paper to perform a full band detection sensitivity study for HyTES, but we do agree this statement was too simplistic. We added some discussion and details from a study by Hall et al. 2008 who looked at the trade-off between gas detection sensitivity and spectral resolution on lines (X-Y):**

*"Of these instruments, HyTES has the highest number of spectral bands (256), which will in general improve the detection sensitivity of most trace gas species, but particularly those gases with sharp spectral features. For example, using a set of ~50 trace gases, Hall et al. (2008) found that species with sharp spectral features such as $H_2S$ and $NH_3$ suffered the greatest sensitivity loss from reduced spectral resolution when simulating the relative sensitivity of data with 64, 32, and 16 spectral channels."*

In addition, when determining detection sensitivity the spectral resolution must be traded against the received radiance per channel. Thus the 256 channels of HyTES result in correspondingly reduced radiometric flux in each spectral channel. This is an important consideration for HyTES in particular, since its QWIP focal plane has a comparatively low quantum efficiency. The statement regarding detection sensitivity should therefore be discussed in greater depth.

**Understood, but the instrument is already built, so the relevance of exploring the effects of spectral resolution versus radiance signal is a moot point in the context of this study. We used QWIPs, since they are radiometrically stable and available in large formats for reasonable cost. QWIP allows us to recalibrate on the hour time scale instead of seconds.**

P2, L36: ": : :provide in situ measurements to validate those results." Explain how this was accomplished.

**We added this sentence to elaborate more on the validation activities:**

*" For example, contemporaneous surface $CH_4$ measurements were made from vehicles with on-board Picarro G2401 or G1301 analyzers while driving along public roads in the domain of HyTES overflights during campaigns over the La Brea tar pits in Los Angeles during 2014, and in the Kern River Oil Field during February 2015."*

P3, L8-10: ": : :relatively low power requirements" with two cryocoolers? Relative to what?

**The power requirements are relatively low compared to the power available on the aircraft we're flying in. The total power requirement is on the order of 1kW. For aircraft this is low, since the ER-2 can support 4kW.**

*"... its small form factor and low power requirement (1 kW) when compared to what the aircraft can support (4kW)."*

P3, L13: "A single sensor calibration is used for an entire field campaign: : :"
TIR sensors are prone to calibration drift and a single calibration is never regarded as valid for even a single flight, let alone an entire campaign. The temperature of the Dewar window alone will change with altitude in the unpressurized airplane cabin. The authors' procedure calls into question the accuracy of their data.

**We should have specified more detail with regard to calibration:**

**For the Twin Otter flights we always calibrate before and after each flight (including any intermediate stops). The AR coatings on the window as well as the low altitude operation has allowed us to get away without an on board calibrator - so far. That being said, a day to day comparison between calibrations in 2014 showed that a single calibration in fact could be substituted with only minor errors for the whole week campaign. The nominal operation**

**for data processing from L0 to L1 is to average the pre and post flight calibrations. We are looking into options for an on-board calibrator (single or two point) for the ER-2 platform, since the thermal gradients may compromise the pre and post flight calibration.**

**This was made clearer in the text:**

*" HyTES is currently configured to fly on the Twin Otter aircraft and Figure 1 shows the aircraft and the HyTES instrument looking nadir in flight. For Twin Otter flights the instrument is calibrated before and after each flight (including any intermediate stops), and the nominal operation for data processing from L0 to L1 is to average the pre and post flight calibrations. That being said, a day to day comparison between calibrations in 2014 showed that a single calibration in fact could be substituted with only minor errors for the whole week's campaign."*

P4, L19: TIR gas detection does not necessarily require a "strong" thermal contrast, but the contrast does have to be finite.

**Changed "strong" to "finite".**

P5, L28-29: Can 100 m really be regarded as "high spatial resolution" in the context of this report?

**Good point. We modified the sentence as follows:**

*"This was verified by comparing these pixels with emissivity information from the ASTER Global Emissivity Database (ASTER GED) at ~100m spatial resolution."*

P7, L5-6: Clarify what is meant by "column" in this context.

**Modified to:**
*".. matrix column-wise fashion (along-track).."*

P7, L13-14: SCR is a dimensionless quantity, which does not comport with the RHS of Eq. (13). Eq. (6) of the Funk et al. (2001) reference provides the correct form of the SCR. This has significant ramifications if Eq. (13) really was used to assess signal strength as the authors state.

**The SCR from eq. 13 is in fact a dimensionless quantity, i.e. for HyTES $q^T$ is 1×256 and b = 256×1, which leads to a dimensionless quantity of size 1×1. This form of the SCR was taken from Theiler (1997). We first started using the formulation in Funk et al. (2001), but on occasion got complex numbers because of negative values within the square root.**

P11, L3-4: "The ability to distinguish between different trace gas signatures within a single plume is a key advantage of TIR hyperspectral data."
As already stated above, this is NOT unique to TIR and therefore cannot be claimed as a specific advantage.

**Please see comment above.**

P15: There are several problems with Table 1:
1. In the header, IFOV should be accompanied by 3 asterisks to link it to the legend at bottom.
**Change made.**

2. For AisaOWL use HgCdTe in place of MCT, as has been done in the remainder of the table.
**Change made.**

3. Both MAKO and SEBASS use Si:As blocked impurity band detectors (and incidentally, "cooled helium prism" refers, albeit in mangled fashion, to a dispersion element, rather than a detector).
**Change made.**

4. The scan angles given for HyTES and SEBASS are approximately twice the actual values.

**Correct, we had total FOV for those two instruments by mistake. Change made.**

An overarching question about this table concerns what the criteria for inclusion were. The SIELETERS group provide a much more comprehensive tabulation of TIR sensors in a paper not cited here (see doi:10.1364/OE.23.016164). The authors' table would benefit from significant revision.

**This was by no means meant to be an exhaustive list of all current airborne sensors, but instead we focused on those that had already been well demonstrated in airborne campaigns.**

**Thank you for the reference. We decided to use this information along with what we already had to create a subset of sensors meeting the following requirements; longwave infrared only, cooled optics, and at least one successful airborne demonstration. Please see the changes in Table 1.**

P17, L18-20: Caption and accompanying figure (P23): At least some of the transmittances shown in Figure 5 do not correspond to the standard atmosphere that the authors claim to have used (in particular, total extinction of the 8.4-micron H2S feature as shown is not credible – see also remarks for Figure 14 on P32, below). It would be instructive to provide the MODTRAN derived column densities for these spectra.

**In order to avoid confusion and inconsistencies with the MODTRAN simulated spectra we decided to plot the normalized Hitran absorption spectra instead for these gases (please see new Figure 5). This is what we originally had plotted in Figure 6 for H2O and CH4 and so we removed that figure since it is now superfluous. The text has been modified in the appropriate locations to take into account these changes.**

P17, L30-33: Caption and accompanying figure (P26): The description of the inset needs rewording. It's said to be a CMF image overlain onto surface temperature, yet the underlying image is actually from Google Earth. The last sentence of this caption only adds to the confusion, but maybe it's moot since the figure is neither cited nor explained in the text and can thus be simply deleted.

**We decided to remove this figure at your suggestion, especially since it was not really conveying any new information that was not already illustrated in other similar figures.**

P32: Figure 14: The peak absorption coefficient for H2S is almost 3 orders of magnitude smaller than ammonia in the TIR band and is unlikely to be detected by this sensor unless its abundance is at potentially lethal levels (a reportable episode at the very least). Has a concentration estimate been computed for this observation?

**No, but our quantitative retrieval for HyTES (Kuai et al. 2016 - AMT in discussion) will be adapted soon to retrieve concentration estimates for H2S. We can only report here what was detected by the CMF as outlined in section 4. We used the same methodology and thresholds for all the gases detected and displayed in this example.**

**We have also detected H2S from HyTES data over the Salton Sea geothermal field on 06/06/2014 shown in the image below, with positive detections shown in yellow. The week following the HyTES overpass, visual inspection of the areas in which $H_2S$ was detected (e.g. section C.1 in figure below) appeared to consist of a combination of decaying fish and dark green/black bubbling decaying bacteria that had a distinctive rotten egg smell indicating the presence of $H_2S$ (W. Johnson, personal communication, 2014). This gives us confidence that what we were detecting was in fact non-lethal levels of H2S.**

[Figure]

Technical Corrections

P2, L22: The HyTES spectral range is everywhere else given as 7.5-12 microns, except here.

**Corrected.**

P5, L15: ISAC is invoked before being defined (P5, L19).

**Some text was moved around to correct this and make this section flow a little better.**

P5, L24-25: The "downwelling" radiances should actually be upwelling.

**Thanks, good catch. Corrected.**

P10, L3-4: It's OK to use SCFH because it's an industry standard, but these values should also be accompanied by their equivalents in SI units.

**We added in the SI units:**

**(5, 10, and 20 kg $CH_4$/hr)**

P10, L13: I spent some time searching for quantitative results in Fig. 12 before realizing that the authors must have meant Fig. 13. Confusing.

**This was made clearer:**

*" We also show results from the HyTES $CH_4$ quantitative retrieval algorithm (Kuai et al., 2016) in Table 2 and Figure 11."*

P11, L4: SO2 is listed twice.

**Corrected.**

P11, L19: Insert "Figure" before "14."

**Corrected.**

P12, L22: "Square Cubic Feet/Hr." should of course be "Standard Cubic Feet/Hr." It's already defined above, so why not just use SCFH? In any case, as mentioned above, these values should be accompanied by their equivalents in SI units.

**Change made:**

*".. small as 250 SCFH (5 kg $CH_4$/hr) at 500 m flight altitude.."*

P13, L6-16, L26-27, L33-38, L48-50, L54-56; P14, L10-12, L15-16, L53-54: Citations incomplete.

**The Endnote template for AMT was interpreting Conference Proceedings entries incorrectly. All corrected and updated.**

P16: Table 2: Provide fluxes in SI units as well as SCFH.

**Change made.**

P24: Figure 6: Y-axis should actually be "Normalized Absorption."

**Change made.**

P31: Figure 13: Provide flow rates in SI units as well as SCFH.

**Change made in legend.**

---

## Referee Comment (RC2) · Anonymous Referee #2 · 9 May 2016

The reviewers did an excellent and detailed set of recommendations. The JPL team did respond to the reviewer recommendations.

I was surprised that it appears JPL did not do an in-house review prior to submission for publication. Thanks to the review comments the article will be an important contribution.